# Laser capture microscopy coupled with Smart-seq2 for precise spatial transcriptomic profiling

Susanne Nichterwitz[1,*], Geng Chen[2,*], Julio Aguila Benitez[1], Marlene Yilmaz[2], Helena Storvall[2,3], Ming Cao[1], Rickard Sandberg[2,3], Qiaolin Deng[2,**] & Eva Hedlund[1,**]

Laser capture microscopy (LCM) coupled with global transcriptome profiling could enable precise analyses of cell populations without the need for tissue dissociation, but has so far required relatively large numbers of cells. Here we report a robust and highly efficient strategy for LCM coupled with full-length mRNA-sequencing (LCM-seq) developed for single-cell transcriptomics. Fixed cells are subjected to direct lysis without RNA extraction, which both simplifies the experimental procedures as well as lowers technical noise. We apply LCM-seq on neurons isolated from mouse tissues, human post-mortem tissues, and illustrate its utility down to single captured cells. Importantly, we demonstrate that LCM-seq can provide biological insight on highly similar neuronal populations, including motor neurons isolated from different levels of the mouse spinal cord, as well as human midbrain dopamine neurons of the substantia nigra compacta and the ventral tegmental area.

[1] Department of Neuroscience, Karolinska Institutet, Retzius v. 8, 171 77 Stockholm, Sweden. [2] Department of Cell and Molecular Biology, Berzelius v. 35, Karolinska Institutet, 171 77 Stockholm, Sweden. [3] Ludwig Institute for Cancer Research, Box 240, 171 77 Stockholm, Sweden. * These authors contributed equally to this work. ** These authors jointly supervised this work. Correspondence and requests for materials should be addressed to Q.D. (email: qiaolin.deng@ki.se) or to E.H. (email: eva.hedlund@ki.se).

Elucidation of gene expression in distinct cellular populations in man and mouse is essential to understand biological processes and disease mechanisms. Scarcity of tissue, especially from patient material, small cellular populations and lack of unique genetic markers requires the use of small amounts of tissue where cells can be readily identified and isolated. Laser capture microscopy (LCM) enables careful dissection of single cells from tissues that have been snap frozen. Sectioning of the tissue thinly, in combination with careful capture of cells presenting a clear nucleus surrounded by cytoplasm ensures that contaminating cells are kept to a minimum. LCM has been coupled with RNA extraction methods to analyse the transcriptome of distinct neuronal populations using microarrays in numerous studies[1–6] and lately also with RNA sequencing[7–9]. LCM of larger tissue cubes in combination with RNA sequencing has also been used for topographical mapping of brain regions[10]. The advantage of LCM to other methods of cell isolation such as fluorescence-activated cell sorting (FACS) is that the positional information of cells is kept without dissociation of tissues and no live cells with genetic labels are needed. Furthermore, tissues from adult and even ageing animals/human brain samples do not easily allow for dissociation followed by RNA sequencing due to the presence of long neuronal processes and a high glial content, but can be analysed instead using LCM-based methods. The number of cells required for LCM coupled with transcriptome profiling has so far often been large with 200–4,000 isolated cells used to retrieve sufficient amounts of analysable RNA[1–3,5–9]. Thus, the technology needed to be further developed to enable efficient sample processing as well as analysis of sparse cell populations in small tissue samples.

Here, we have developed LCM-seq that couples LCM with the Smart-seq2 RNA sequencing technology[11,12] for robust and efficient sequencing of polyA$^+$ RNA in neurons isolated from mouse and human tissues. By optimizing multiple steps in the procedure, including direct lysis of cells without performing RNA extraction, we can now acquire high-quality RNA sequencing data down to single LCM dissected cells.

## Results

**Improving tissue preparation and scaling down cells captured.** We sectioned tissues at 12 μm and visualized motor neurons (MNs) in lightly fixed tissue sections using a Histogene (Arcturus) quick staining (Fig. 1; Supplementary Fig. 1a–d). Usually this procedure requires the use of a costly commercial staining kit, which includes high percentage ethanol solutions and xylene. However, we discovered that use of the kit was not necessary, as the Histogene staining solution could be combined with regular (off-the-shelf) ethanol (99.7%). We also omitted the last dehydration step, which requires toxic xylene and instead only used ethanol fixation. Standard methods use RNA isolation kits to extract RNA from fixed, isolated cells. We found that we could retrieve RNA extraction, by direct lysis of isolated cells in a mild hypotonic solution, as for live single-cell preparations[12]. All these improvements significantly reduced costs and improved the efficiency of the procedure. We named this improved method LCM-seq (Laser capture microscopy coupled with Smart-seq2). Generally when LCM is coupled with downstream global transcriptome analyses, several hundred, if not thousands, of cells are needed to ensure high quality of data. As some cellular populations are very small and tissue often in scarcity, we aimed to carefully document the lowest number of cells needed to retrieve high-quality libraries. We started with capturing 120 cervical spinal MNs at postnatal day 5 (P5) and subsequently scaled down to 50 cells, 30 cells, 10 cells, 5 cells, 2 cells and finally 1 cell. We also included an experimental group where 120 cells were isolated by LCM and then subjected to RNA extraction before cDNA synthesis. To compare cDNA yield after direct lysis followed by reverse transcription, we performed 18 cycles of PCR amplification for all samples and measured the cDNA quantity and quality with an Agilent 2100 Bioanalyzer using a high sensitivity DNA kit. For the direct lysis of samples the entire lysis volume was used for cDNA library preparation, while for the RNA extraction samples half the volume was used because of volume restrictions of the protocol (Table 1). As expected the total cDNA yield decreased when fewer cells were captured, but the cDNA profiles were of comparable quality for all cell numbers. The cDNA yield from direct lysis of 120 cells ($147.4 \pm 15.7$ ng, adjusted $P = 0.0081$ ANOVA, Dunnett's multiple testing correction) was significantly higher than that of the RNA extraction group ($39.7 \pm 11.9$ ng). Direct lysis from 50, 30 and 10 cells resulted in a comparable cDNA yield to the RNA extraction group (adjusted $P = 0.1611$, $P = 0.9997$ and $P = 0.9798$, respectively, ANOVA, Dunnett's multiple testing correction) demonstrating the efficiency of our direct lysis method (Supplementary Fig. 1e,f). The total area of collected cells (μm$^2$), yield of total cDNA (ng), the number of replicates conducted for each group and the total mapping ratio is listed in Table 1. Our data show that transcriptome sequencing from single MNs is now possible at a high success rate, 62% for 1 cell, 81% for 2 cell and 100% from 5 to 120 cell samples, using LCM-seq.

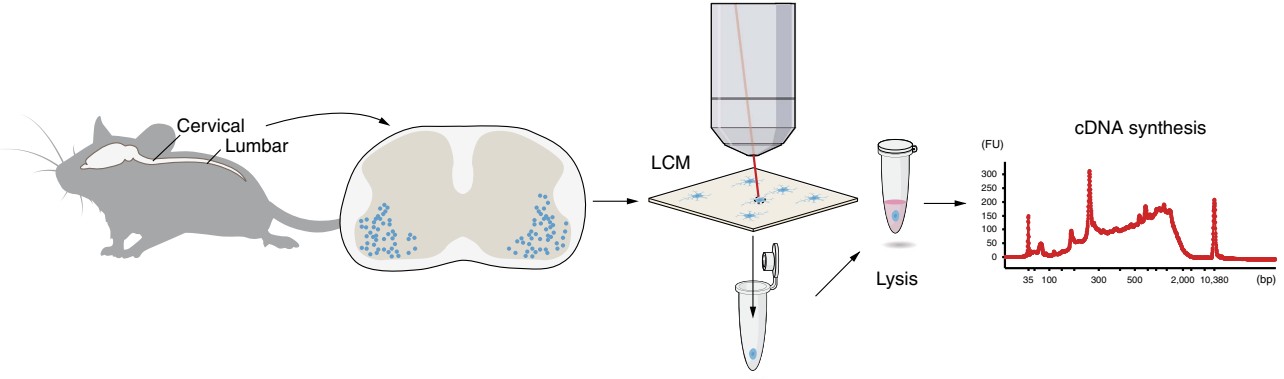

**Figure 1 | Schematics of Smart-seq2 coupled with LCM (LCM-seq) for analysis of single neurons.** Spinal cords isolated from mouse pups were sectioned at 12 μm thickness at cervical and lumbar levels. MNs were visualized using Histogene staining, 120, 50, 30, 10, 5, 2 and 1 cell captured by a Leica LMD7000 system and collected in PCR tubes. Cells were lysed, cDNA synthesized, amplified and analysed using a Bioanalyzer and sequencing libraries subsequently prepared. The cDNA profile is exemplified by a Bioanalyzer profile of 50 captured MNs. bp, basepairs; FU, fluorescent units.

| Table 1 \| Specification of mouse motor neuron samples and corresponding cDNA libraries. | | | | | | |
|---|---|---|---|---|---|---|
| Group | N | | Cells collected | Total area (µm$^2$) | cDNA yield (ng) | Total mapping ratio (%) |
| | Animals | Samples | | | | |
| RNA extract | 5 | 5 | 120.8 ± 4.0* | 59,562 ± 1,204 | 39.7 ± 11.9 | 93.7 ± 0.8 |
| 120 | 4 | 4 | 117.5 ± 6.2 | 61,661 ± 3,465 | 147.4 ± 15.7 | 89.8 ± 0.7 |
| 50 | 5 | 8 | 53.9 ± 0.8 | 29,195 ± 691 | 87.1 ± 19.0 | 90.4 ± 0.6 |
| 30 | 5 | 9 | 32.2 ± 1.4 | 18,363 ± 822 | 43.3 ± 10.2 | 89.3 ± 0.9 |
| 10 | 5 | 10 | 10.4 ± 0.2 | 6,022 ± 202 | 25.0 ± 3.0 | 89.5 ± 0.6 |
| 5 | 5 | 8 | 5.4 ± 0.2 | 2,774 ± 304 | 7.2 ± 1.1 | 85.3 ± 1.7 |
| 2 | 4 | 13 | 2 ± 0 | 1,272 ± 127 | 4.9 ± 0.7 | 79.9 ± 1.5 |
| 1 | 5 | 13 | 1 ± 0 | 719 ± 39 | 6.0 ± 1.2 | 79.0 ± 1.1 |
| 0 | NA | 5 | NA | NA | 0.4 ± 0.1 | NA |

NA, not applicable.
$N$ = number of animals and samples isolated from these; total cDNA yield (ng) as measured on the Bioanalyzer in the 100–9,000 bp range; total mapping ratio (%) to the mouse genome mm10.
Values are mean ± s.e.m.
*Half of the eluted volume after RNA extraction of 120 MNs was used for cDNA library preparation, which corresponds to a cDNA yield from approximately 60 cells.

**LCM-seq is reproducible and sensitive down to single cells**. Next, we wanted to compare the performance of LCM-seq using direct lysis with LCM-seq coupled with a standard RNA extraction protocol. To do this, we cross-compared the transcriptome profiles of all samples including RNA extraction and direct lysis of 120 cells down to 1 cell (all cervical MNs from P5) subjected to LCM-seq. To ensure that all samples analysed were indeed spinal MNs, especially pertinent for 1–5 cell samples, we confirmed that a number of MN markers, including the transcription factors *Islet-1*, *Islet-2* and *Mnx1*, and choline acetyl transferase (*Chat*), neurofilament heavy chain (*Nefh*) and peripherin (*Prph*) were expressed in these samples (Supplementary Fig. 2a). Furthermore, to evaluate the level of glial contamination in our samples we also analysed the expression of the glial markers *Gfap*, *Mfge8*, *Aif1*, *Cx3cr1*, *Gpr17*, *Itpr2* and *Cnksr3* and compared with glia samples from a recently published paper[13] (GEO accession number GSE52564). This clearly demonstrates that the isolated cells express high levels of MN markers, while they did not express or expressed very low levels of some glial markers (Supplementary Fig. 2a). Thus, LCM results in the isolation of highly enriched MN samples.

Strikingly, LCM-seq significantly improved the reproducibility of gene detection reflected by the proportion of genes with low and medium expression levels commonly detected across replicates compared with the standard RNA extraction protocol, and this reproducibility remained equally good even down to 10 cells (Fig. 2a). The 1 and 2 cell samples subjected to LCM-seq showed the lowest reproducibility, which partially reflected the biological variability among MNs. Moreover, our LCM-seq protocol reduced the technical variation for genes expressed at low and medium levels in samples with 120–30 cells compared with that of samples subjected to the standard RNA extraction approach (Fig. 2b). The mean number of detected genes in samples with 120 cells subjected to LCM-seq was higher (1,743 more genes for the 0.1 RPKM (reads per kilobase of transcript per million mapped reads) threshold, $P = 0.04$, Student's *t*-test) than that of the RNA extraction protocol samples (Fig. 2c). The mean number of detected genes in the samples subjected to RNA extraction was comparable to that of 30 and 10 cell samples sequenced using LCM-seq. The 5, 2 and 1 cell samples showed lower mean numbers of detected genes (Fig. 2c). However, a comparison to live single spinal MNs derived from mouse embryonic stem cells (mESCs) sequenced by Smart-seq2 (Supplementary Fig. 2b) demonstrated a comparable number of detected genes between LCM-seq MNs and live single MNs (Supplementary Fig. 2c). Subsequently, we evaluated the 5′ to 3′ coverage bias of RNAs in MNs subjected to LCM-seq.

We performed analyses for transcripts with the length of < 3 kb (62.5%), 3–10 kb (36.3%) and > 10 kb (1.2%) and compared with live single mESC-derived MNs. Live single mESC-derived MNs sequenced by Smart-seq2 displayed a better gene body coverage than LCM-seq samples, indicating that the LCM procedure, with its associated tissue dissection, cryostat sectioning and staining, induces partial degradation of RNAs (Supplementary Fig. 3a–c). While a higher number of genes could be detected using our direct lysis approach (Fig. 2c), the RNA extraction method showed a slightly better coverage at 5′ of transcripts with length < 3 kb (Supplementary Fig. 3a). This could be due to that the RNA extraction method has a purification step, which can cause a loss of RNAs, but at the same time improves the efficiency of reverse transcription for the remaining transcripts.

Spearman's correlation of all expressed genes within the same groups and across different groups was high (Fig. 2d). We found that the gene expression profiles of 120 cell samples correlated highly with 50, 30 and 10 cell samples. When reducing the number of cells from 10 to 1, gene expression correlation across distinct replicates gradually decreased due to cell to cell variability and technical noise, but was just slightly lower than other groups (Fig. 2d). Principal component analysis (PCA) of the different cell number groups based on the top 500 variable genes in expression confirmed the high similarity between the RNA extraction group and the 120 cell direct lysis group (Fig. 2e). Moreover, 50, 30 and 10 cell groups subjected to LCM-seq also showed close clustering with that of 120 cell direct lysis group and the RNA extraction group. This data implies that isolation of 10 MNs could represent the diversity among MNs in the cervical spinal cord of P5 mice. This was further supported by PCA where 120 and 10 cervical MN groups clustered together, away from 120 lumbar MN groups (Supplementary Fig. 2d). The disperse clustering among 1 and 2 cell samples agreed with the slightly lower expression correlation (Fig. 2e).

Finally, we wanted to investigate the cell size limit of LCM-seq by isolating cells with significantly smaller areas than the spinal MNs (areas of 500–750 µm$^2$) analysed so far. Consequently, we dissected MNs from the hypoglossal nucleus with cell soma areas of 200–300 µm$^2$, and from the dorsal motor nucleus of vagus (DMX) with cell soma areas of 130–200 µm$^2$ (Supplementary Fig. 4a–c). LCM-seq of single cells and 5-cell pools from these two anatomical nuclei demonstrated that cells with smaller areas can be successfully sequenced using this method. Single-cell samples contained an average of 4,958 detectable genes, while 6,945 detectable genes were identified in 5-cell sample pools (Supplementary Fig. 4d). The mean percentage of mapped reads was 75.13% for the single neurons and 78.43% for the 5-cell

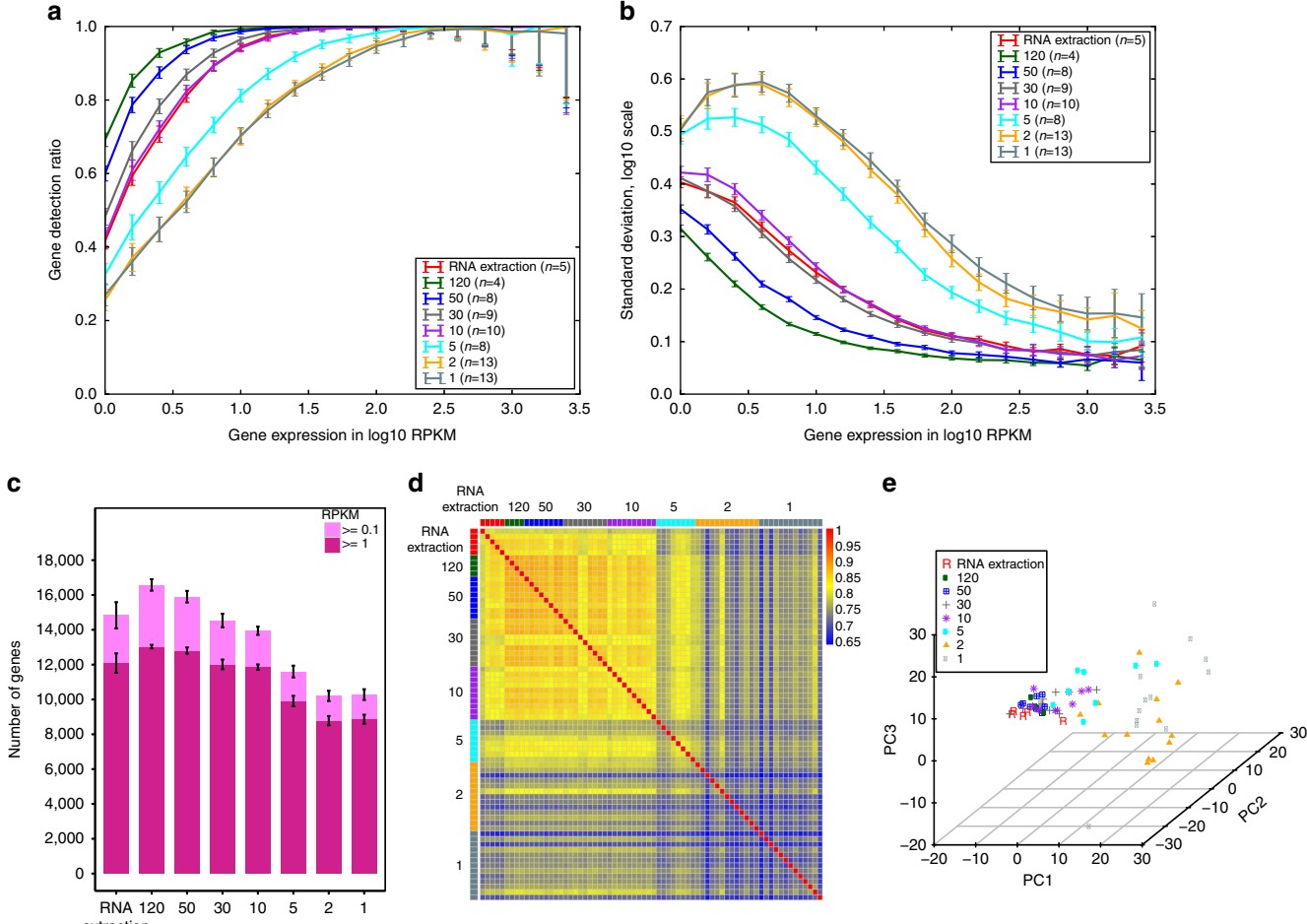

**Figure 2 | LCM-seq improved the sensitivity of gene detection and is applicable to single cells.** (**a**) Reproducibility of detected genes was evaluated by comparing all possible pairwise comparisons within replicates of each group (control RNA extraction and 120, 50, 30, 10, 5, 2 and 1 LCM-seq samples), shown as mean with 90% confidence interval of reproducible ratios. LCM-seq improved reproducibility of detection of low and medium level expressed genes compared to the RNA extraction protocol (RNA extraction) (mean ± s.e.m., $n \geq 4$). (**b**) Standard deviation of gene expression within replicates binned according to gene expression levels. LCM-seq samples of 120–30 cells showed reduced technical variation for low and medium level expressed genes compared with the RNA extraction protocol (RNA extraction) (mean ± s.e.m., $n \geq 4$). (**c**) Mean number of genes detected in the different sample groups (mean ± s.e.m.). A larger number of genes were detected in the 120 cell samples subjected to LCM-seq compared to the group subjected to RNA extraction before sequencing (0.1 RPKM as cutoff, $P = 0.04$, Student's $t$-test). (**d**) Gene expression correlation was high for 120, 50, 30 and 10 cell samples, while 5, 2 and 1 cell samples showed the heterogeneity among spinal MNs (Spearman's correlation, genes expressed $\geq 1$ RPKM in at least one sample were used). (**e**) PCA for all groups based on top 500 variable genes in expression confirmed the high similarity between 120 and the RNA extraction group as well as with 50, 30 and 10 cell LCM-seq samples.

samples (Supplementary Fig. 4e, displayed as mean ± s.e.m.). In summary, our LCM-seq protocol significantly improved the reproducibility and sensitivity of gene detection, and is applicable to single neurons.

**LCM-seq reveals distinct spinal MN transcriptome profiles**. To show the validity of LCM-seq and evaluate if neuronal subpopulations with highly similar functions could be clearly distinguished, we compared transcriptome profiles of MNs isolated from cervical spinal cord (cSC; fore limb innervating) and lumbar spinal cord (lSC; hind limb innervating) of P5 mice, using ~120 cells per sample to best represent the complexity of each neuronal population. PCA of cSC and lSC MNs, based on the top 500 variable genes, showed that these two neuronal populations were clearly separated along the first principal component (PC1, 51% variance) (Fig. 3a). Thus, spinal MNs displayed unique identities based on their positions along the

anterior–posterior axis of the spinal cord, which could be revealed by LCM-seq. Further comparison of cSC and lSC MNs revealed that 899 genes were differentially expressed between the two groups (Fig. 3b, adjusted $P < 0.05$, Wald test Supplementary Data 1, see Methods section). Gene ontology (GO) enrichment analysis suggested that the differentially expressed genes were mainly involved in synaptic transmission, localization, organization of biogenesis, cell-to-cell signalling, development, metabolic processes, neurotransmitter transport and neuro-muscular processes (adjusted $P < 0.01$, Fisher's exact test, topGO, Supplementary Fig. 5, see Methods section).

Homeobox transcription factors (Hox) confer positional identity to the segmented, developing hindbrain and spinal cord and are expressed in a colinear fashion with distinct anterior borders and diffuse posterior borders[14]. We found that cSC and lSC MNs showed distinct Hox gene expression profiles. Specifically, 12 Hox genes were differentially expressed, namely, *Hoxa11, Hoxa10, Hoxc10, Hoxd10, Hoxa9, Hoxd9, Hoxc8, Hoxa7,*

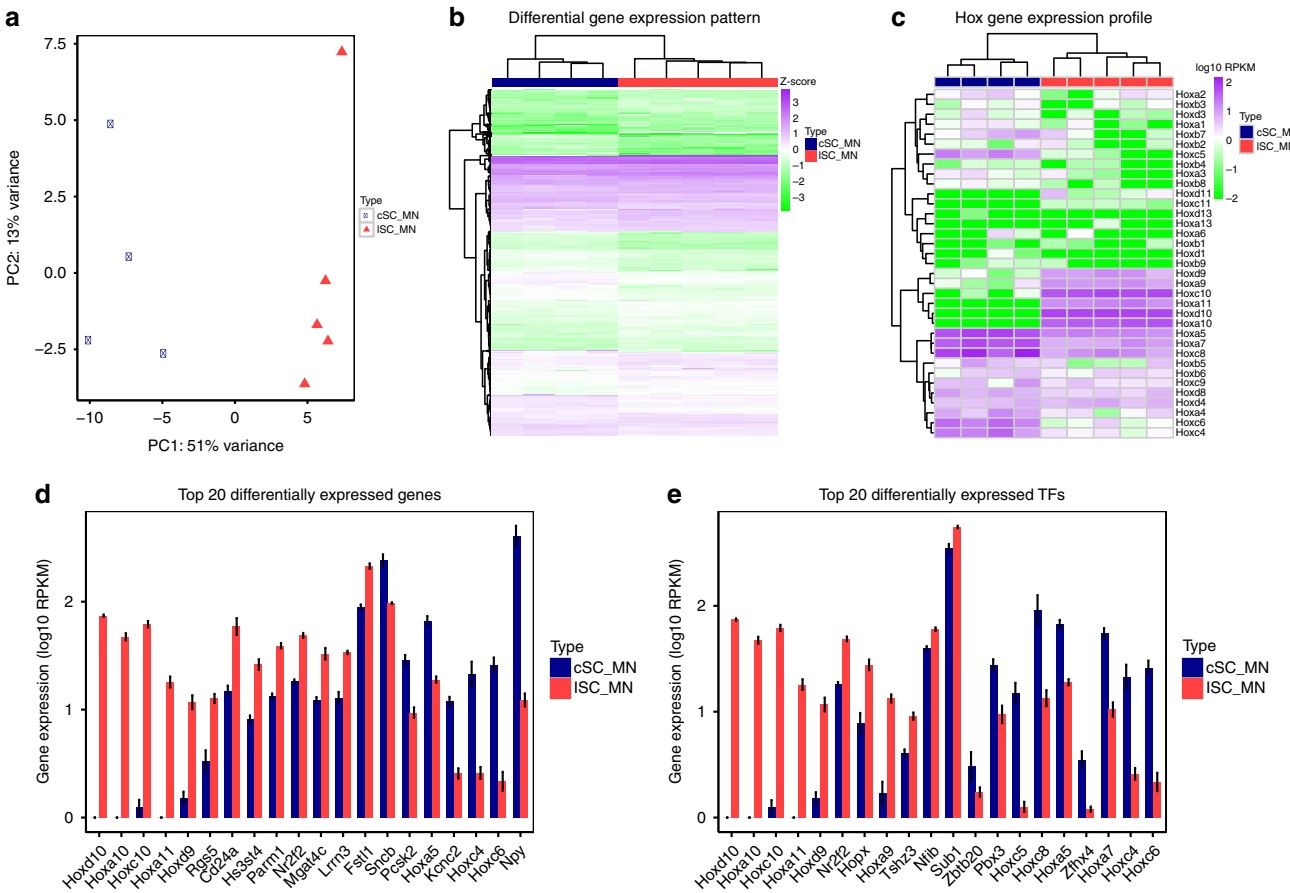

**Figure 3 | LCM-seq revealed the unique identities of cervical and lumbar spinal MNs.** (**a**) PCA for cSC ($n = 4$ mice) and lSC ($n = 5$ mice) MNs (120 neurons/sample) based on the top 500 variable genes in expression showed that the two groups separated along the 1st component. (**b**) Genes (899) were differentially expressed between cSC MNs and lSC MNs (adjusted $P < 0.05$, Wald test). Heatmap shows the Z-score of each gene, which was calculated based on $log_{10}$-transformed RPKM values. (**c**) cSC MNs and lSC MNs showed unique Hox gene expression profiles. (**d**) The top 20 differentially expressed genes between cSC and lSC neurons included eight Hox genes, and for example, *Npy*, *Fstl1* and *Sncb* (sorted by fold change and adjusted *P*, shown as mean ± s.e.m.). (**e**) The top 20 differentially expressed transcription factors (TFs) between cSC and lSC neurons included twelve Hox genes and for example, *Nr2f2* and *Zfhx4* (sorted by fold change and adjusted *P*, shown as mean ± s.e.m.).

*Hoxc6, Hoxa5, Hoxc5* and *Hoxc4* (Fig. 3c). Eight of these 12 Hox genes were among the top 20 differentially expressed genes (Fig. 3d). Thus, the Hox gene profile also defined specific MN populations during postnatal development. Neuropeptide Y (*Npy*) and the voltage-gated potassium channel *Kcnc2* also showed prominent differences between the two MN groups with higher expression levels in cSC MNs. Regulator of G protein signalling 5 (*Rgs5*), *Cd24a* and Follistatin-related protein 1 (*Fstl1*) showed the opposite expression pattern with a higher level in lSC MNs (Fig. 3d). Evaluation of the top 20 differentially expressed transcription factors (TFs), using the AnimalTFDB 2.0 database of Zhang *et al.*[15], between cSC and lSC MNs revealed an inclusion of all 12 differentially expressed Hox genes identified (see above) and eight other TFs including *Nr2f2, Hopx, Tshz3, Nfib, Sub1, Zbtb20, Pbx3* and *Zfhx4* (Fig. 3e). Consequently, our results demonstrated that LCM-seq can be applied to distinguish closely related neuronal populations and that cSC and lSC MNs display large gene expression differences, despite their similar functions, which are closely related to their positional identity.

**LCM-seq reveals specific transcriptomes of human neurons.** Subsequently, we investigated if LCM-seq could be applied to human post-mortem samples that are usually partially degraded due to long tissue processing times (Supplementary Table 1).

We isolated spinal MNs ($179.7 ± 30.8$ MNs per sample) and midbrain dopamine (mDA) neurons from the ventral tegmental area (VTA; $169.3 ± 17.6$ cells per sample) and substantia nigra pars compacta (SNc; $157.0 ± 11.5$ cells per sample) using Histogene staining (Supplementary Fig. 6a–h). Assuringly, RNA could be successfully retrieved from human post-mortem samples using LCM-seq (Supplementary Fig. 6m–o) with a mean 90.53% of sequenced reads that could be mapped to the human reference genome hg38 across 13 human samples (Table 2). Moreover, these samples contained an average of 10,718 genes expressed at $≥1$ RPKM (14,893 genes expressed at $≥0.1$ RPKM) (Fig. 4a). Clustering of spinal MNs and mDA neurons based on the top 500 variably expressed genes showed a clear separation of these two neuronal populations (Fig. 4b). Further analysis revealed that 4,903 genes were differentially expressed between two types of neurons, among which 290 were TFs (adjusted $P < 0.05$, Wald test, Supplementary Data 2). Among the top 20 differentially expressed genes, dopamine transporter (*DAT*; *SLC6A3*), tyrosine hydroxylase (*TH*), *HMP19* and *SLC2A13* (facilitated glucose transporter) were more highly expressed in mDA neurons. Sixteen of the top 20 differentially expressed genes were preferentially expressed in spinal MNs, for example, Leucine zipper protein 1 (*LUZP1*), PRUNE homologue of 2 (*PRUNE2*), clusterin (*CLU*), SPARC-like protein 1 (*SPARCL1*), neurofilament medium polypeptide (*NEFM*) and heavy polypeptide (*NEFH*),

**Table 2 | Details of human neuron samples and the corresponding LCM-seq libraries.**

| Group | N | | | Total area (μm²) | cDNA yield (ng) | Total mapping ratio (%) |
|---|---|---|---|---|---|---|
| | Cases | Samples | Cells isolated | | | |
| Spinal MNs[H] | 4 | 6 | 179.7 ± 30.8 | 96,626 ± 30,057 | 24.4 ± 9.7 | 85.0 ± 1.0 |
| mDA SNc[H] | 3 | 4 | 157.0 ± 11.5 | 86,216 ± 21,474 | 114.7 ± 28.7 | 92.9 ± 0.3 |
| mDA VTA[H] | 3 | 3 | 169.3 ± 17.6 | 82,198 ± 28,012 | 72.0 ± 25.7 | 93.7 ± 0.4 |
| mDA[qTH] (SNc + VTA) | 3 | 5 | 191.8 ± 8.0 | 59,512 ± 8,372 | 42.7 ± 11.7 | 92.2 ± 0.5 |

H, histogene staining; mDA, midbrain dopamine neurons; MNs, spinal motor neurons; qTH, quick tyrosine hydroxylase antibody staining; SNc, *Substantia nigra compacta*; VTA, ventral tegmental area.
Values are mean ± s.e.m.

nuclear factors I/C (*NFIC*) and I/A (*NFIA*), potassium channels (*KCNA1* and *KCNMB4*), S100 calcium-binding protein A10 (*S100A10*) and sushi domain-containing protein 2 (*SUSD2*) (Fig. 4c). Human spinal MNs also expressed a number of Hox genes, which confer positional identity during development (Supplementary Fig. 7a). The differentially expressed genes between spinal MNs and mDA neurons were enriched in a number of biological processes including synaptic transmission, cell–cell signalling, localization, nervous system development, transmembrane transport, synapse organization and neuro-transmitter transport (adjusted $P < 0.01$, Fisher's exact test, topGO, Supplementary Fig. 7b).

MNs can be readily isolated from mouse and human tissues using only a histological staining as these cells are easily distinguished based on their location and size. mDA neurons can also be readily identified in human tissue as they contain neuromelanin (Supplementary Fig. 6f,g). However, as we wanted our method to be applicable also for neurons that cannot readily be distinguished and require an antibody staining before LCM, which increases the risk for RNA degradation, we compared mDA neurons stained using the histological staining (Histogene) or an antibody-directed against tyrosine hydroxylase (TH) (Supplementary Fig. 6i–l,o). Here, we isolated mDA neurons both from the SNc and the VTA. We found that reliable sequencing quality could be achieved for samples subjected to either Histogene staining (mean 93.3% mapping ratio to genome, $n = 5$) or quick TH antibody staining (mean 92.2% mapping ratio to genome, $n = 5$). Moreover, both mDA neuron groups yielded a mean of 11,420 genes with $\geq 1$ RPKM (a mean of 15,850 for 0.1 RPKM cutoff). However, a larger number of genes (912 more genes for 0.1 RPKM as cutoff, $P = 0.03$, Student's *t*-test) could be detected in Histogene samples compared with TH antibody staining samples (Fig. 4d). Nonetheless, the expression of mDA neuron markers including *DAT*, *EN1*, *EN2*, *FOXA2*, *LMX1B*, *OTX2*, *PITX3* and *TH* remained comparable (Fig. 4e). To evaluate the consequences of increased staining duration on LCM-seq quality we prolonged our standard primary TH antibody incubation time from 4 min to 20 and 60 min, and subsequently captured SNc mDA neurons. Importantly, while the mapping ratio decreased slightly with increased staining time (Supplementary Table 2), we found that the number of detected genes in the three different 'primary antibody staining time' groups were comparable, as was mDA marker gene expression (Supplementary Fig. 8a,b). Thus, LCM-seq can be applied to longer staining times.

Notably, although both staining protocols coupled with LCM-seq generated reliable data it is important to use only one of the methods for a given study, so not to confound biological variation with technical differences caused by the specific staining method. We further analysed if LCM-seq could be applied to distinguish SNc and VTA, two close subtypes of mDA neurons, isolated after Histogene staining. Indeed, hierarchical clustering based on the top 500 variable genes showed that SNc and VTA

mDA neurons clustered separately from each other (Fig. 4f). Differential expression analysis revealed that 111 genes were differentially expressed between SNc and VTA neurons, where for example, *CALBINDIN 1* (*CALB1*), *SLIT3* and *WNT5A* were more highly expressed in VTA, while *SOX6*, *VAV3*, *ATP2A3*, *SLIT1* and *WNT3* showed higher expression levels in SNc mDA neurons (adjusted $P < 0.05$, Wald test, Fig. 4g; Supplementary Data 3). The differentially expressed genes between SNc and VTA mDA neurons were enriched in the biological processes of, for example, regulation of localization, cell–cell signalling, nervous system development (Supplementary Fig. 7c). In conclusion, LCM-seq is a robust method for obtaining reliable transcriptome sequencing even from poor quality frozen tissues, such as human post-mortem brain and spinal cord tissues.

**LCM-seq is applicable to single cells in human tissues**. We finally investigated if LCM-seq could be applied to a low number of cells isolated from human post-mortem CNS tissues. We compared larger cell pools containing $179.7 \pm 30.8$ MNs/sample (bulk) and subsequently scaled down to 10 MNs, 5 MNs and finally 1 MN, which were subjected to direct lysis and LCM-seq. Gene expression analysis demonstrated that these human MNs expressed the MN markers *ISLET-1/2*, *MNX1*, *CHAT*, *NEFH* and *PRPH* (Fig. 5a). A large number of expressed genes were detected in all cell groups, and even the single MN group had a mean of 7,655 genes expressed at $\geq 1$ RPKM (Fig. 5b). Moreover, the Spearman's correlations of gene expression among bulk, 10, 5 and 1 MN groups were $> 0.6$ (Fig. 5c). PCA clustering of the MN groups based on the top 500 variable genes demonstrated that the MN groups clustered well together and away from the human mDA neuron samples (bulk), isolated from human post-mortem tissues (Fig. 5d). Thus, our results demonstrate that LCM-seq can be applied even to single human MNs isolated from partially degraded post-mortem tissues.

## Discussion

RNA sequencing has increasingly become a method of choice for profiling gene expression of targeted cell types. Lately single-cell RNA sequencing techniques have further advanced this application to rare and scarce cell populations. However, for unbiased identification of rare cell populations by single-cell RNA sequencing a very large amount of cells need to be collected and analysed due to the lack of proper genetic markers[16].

In this study we have developed LCM-seq, an efficient and robust method that combines LCM with Smart-seq2 and can be applied to very low cell numbers including single cells from both mouse and human tissues. This method provides an alternative solution to resolve transcriptome profiles of rare cell populations at known positions at much lower cost as fewer cells are needed. By avoiding RNA extraction and using off-the-shelf reagents, LCM-seq is easier and cheaper to implement and significantly

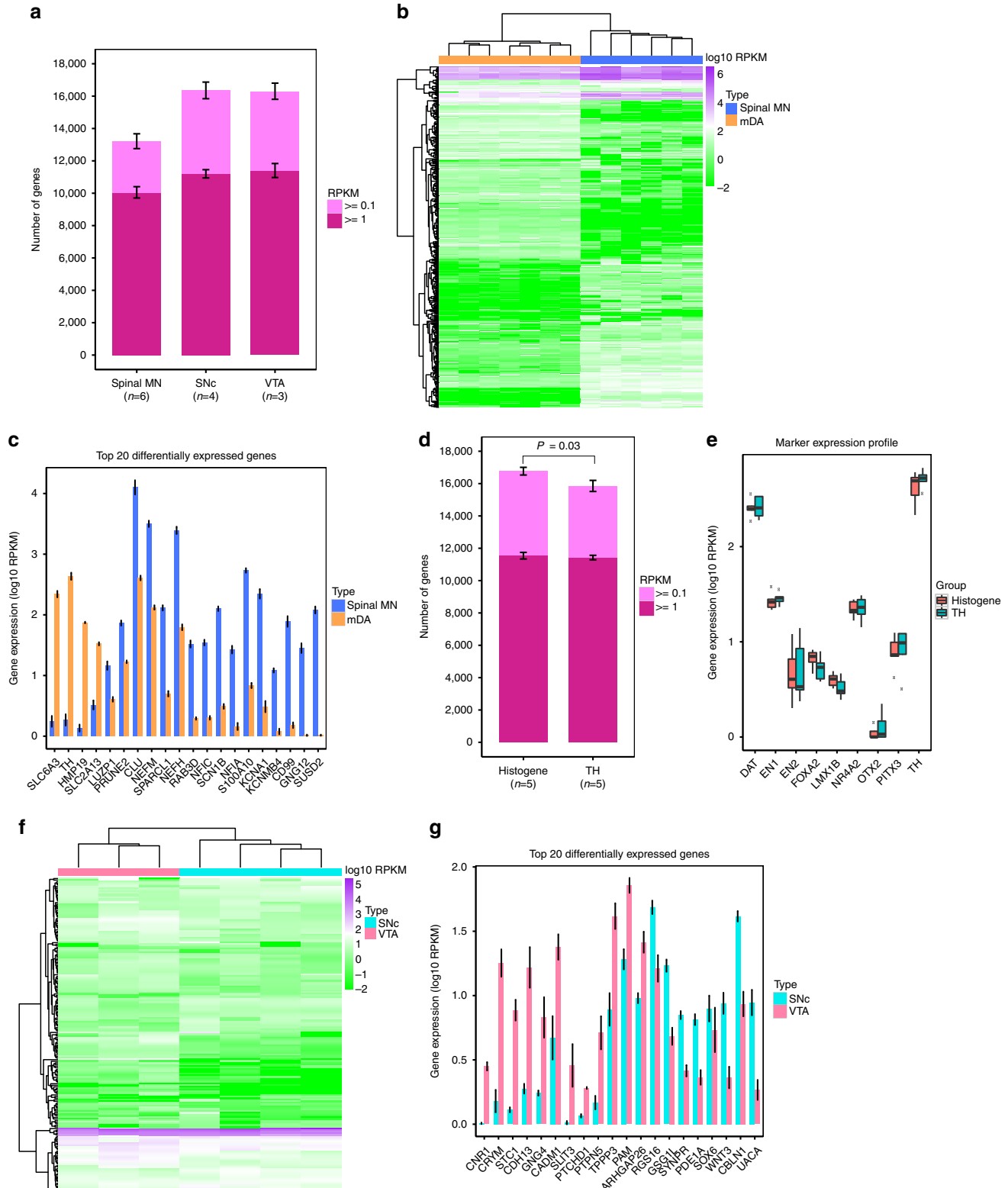

**Figure 4 | LCM-seq is applicable to partly degraded human post-mortem tissues and reveals differential gene expression of spinal MNs and mDA neurons.** (**a**) Mean number of genes detected in MNs ($n = 6$) and mDA neurons (SNc: $n = 4$; VTA: $n = 3$, mean ± s.e.m.). (**b**) Clustering of MNs ($n = 6$) and mDA neurons ($n = 7$) based on the top 500 variable genes in expression. (**c**) Top 20 differentially expressed genes between spinal MNs and mDA neurons (adjusted $P < 0.05$, Wald test, DESeq2, sorted by fold change and adjusted $P$, mean ± s.e.m.). (**d**) A larger number of genes was detected in mDA neurons using Histogene staining than the quick TH antibody staining ($P = 0.03$, Student's $t$-test at 0.1 RPKM cutoff, mean ± s.e.m.). (**e**) Comparable mDA neuron marker expression levels were detected between the Histogene and quick TH staining groups. Boxes range from the 25th to the 75th percentile, with the centerline representing the 50th percentile. Outliers are shown as dots. (**f**) Clustering of SNc ($n = 4$) and VTA ($n = 3$) neurons, using Histogene staining, based on the top 500 variable genes in expression. (**g**) The top 20 differentially expressed genes between SNc and VTA neurons (adjusted $P < 0.05$, Wald test, DESeq2, sorted by adjusted $P$, mean ± s.e.m.).

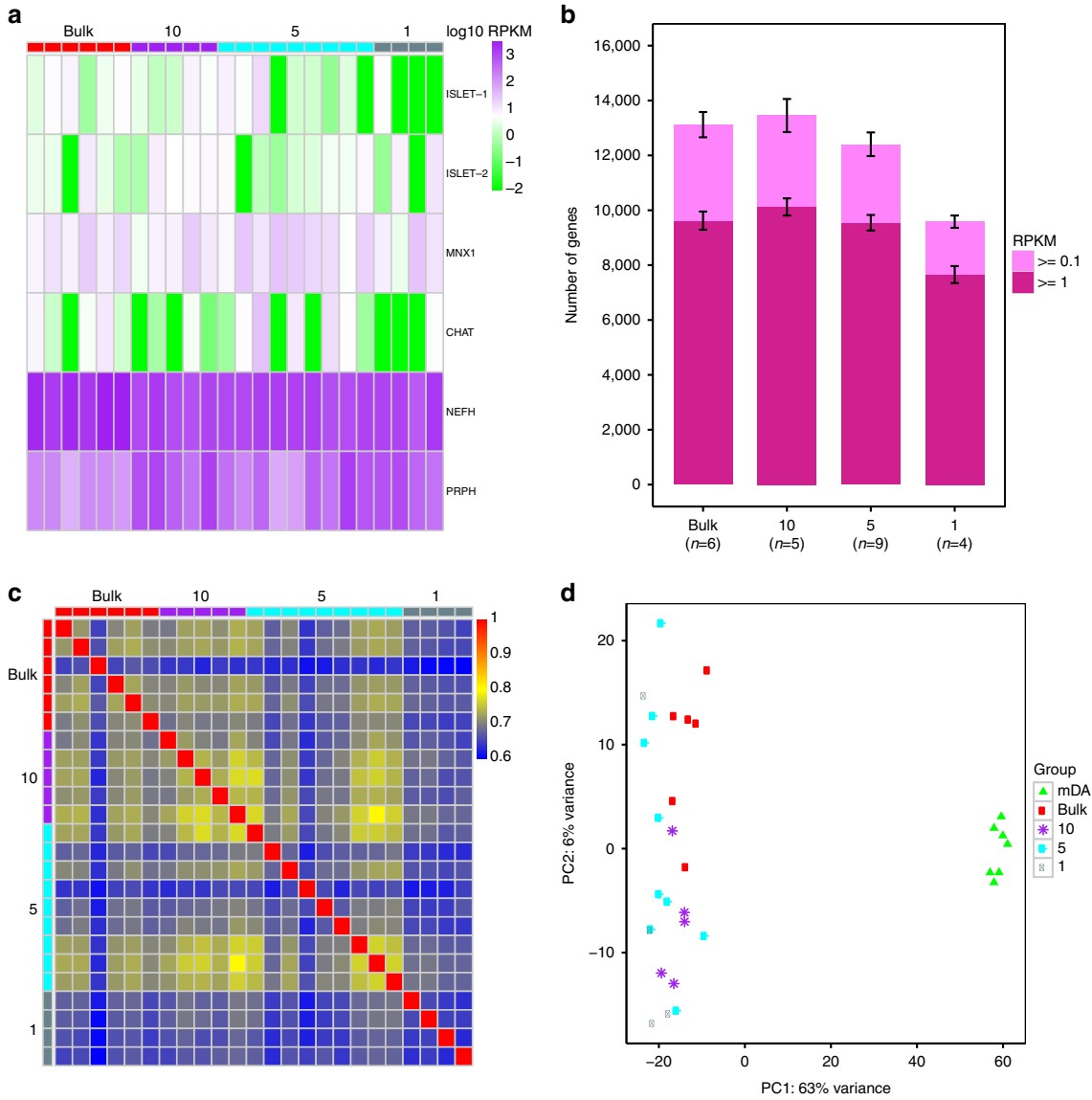

**Figure 5 | LCM-seq is applicable to single human MNs isolated from post-mortem tissues. (a)** Human MNs expressed the MN-specific markers *ISLET-1/2, MNX1, CHAT, NEFH* and *PRPH*. **(b)** Mean number of genes detected in the distinct groups of different cell numbers; bulk (179.7 ± 30.8 cells), 10, 5 and 1 MN (displayed as mean ± s.e.m.). **(c)** Gene expression correlation for bulk, 10, 5 and 1 cell samples (Spearman's correlation; genes expressed at ≥1 RPKM in at least one sample were used). **(d)** PCA of bulk, 10, 5 and 1 MN groups and bulk mDA neurons, based on the top 500 variable genes.

improves reproducibility and sensitivity of gene detection compared with previously used methods. By scaling down the number of captured cells, scarce tissues and cell populations can be analysed and we show that 10 MNs display comparable gene expression profiles to samples containing more (30–120) cells. Importantly, our method could also be applied to single LCM dissected cells, which enables the possibility to study cell-to-cell heterogeneity in frozen tissues. By analysing our samples for neuronal and glial markers we could also demonstrate that LCM results in the isolation of enriched pure neuronal populations. Strikingly, our method can also be used down to the single-cell level on human post-mortem tissues that undergo longer processing times and thus are partially degraded. Furthermore, LCM-seq was compatible with antibody staining, which renders it equally useful on less distinguishable cell types of interest.

Our analysis of cervical and lumbar spinal MNs revealed a number of important differences between these neurons that were previously unknown. For example, *Nr2f2* was previously

shown to be highly expressed in spinal MNs[17], but no axis information was available. The TFs *Pbx3* and *Nfib* had not previously been implicated in motor neuron function, but our data showed their presence in MNs and differential expression along the spinal cord (as confirmed also by Allen Brain Atlas). Furthermore, our analysis revealed that cervical and lumbar MNs maintain unique Hox gene codes postnatally. The combinatorial expression of multiple Hox genes during development are known to give spinal MNs their specific identities and regulate connectivity with muscle targets[18–21]. Our data suggests that these specific Hox codes are not only important for developmental processes to take place, but also to maintain MN identity and function throughout life both in mouse and man. This continued expression of Hox genes after embryonic development is in accordance with our previous finding in cervical MNs of adult rats[1]. In conclusion, spinal MNs showed unique identities, despite their similar functions, based on their position along the anterior–posterior axis of the spinal cord, which could be revealed by LCM-seq. Finally, our gene expression

analysis could be highly relevant for future studies on MN function and for directed generation of specific MN populations from stem cells.

Importantly, by comparing the transcriptome profiles of SNc and VTA mDA neurons, we identified 111 differentially expressed genes, which could aid in deducing underlying mechanisms of their differential vulnerability to degeneration in Parkinson's disease[22,23]. Here, we revealed a number of novel differences in *CDH13, WNT3, WNT5A* and *SLIT1* gene expression, while confirming several genes previously shown to be differentially expressed in SNc and VTA in mouse or rat, including *CALBINDIN 1 (CALB1)*[24–26], which was highly expressed in VTA and *SOX6* (refs 2,27), *CBLN1* (ref. 2), *VAV3* (ref. 2) and *ATP2A3* (ref. 2) that were more prominent in SNc.

In summary, we demonstrated the advantage of LCM-seq in analysing the transcriptome of neurons in mouse and human post-mortem tissues. Our approach could be broadly used for analysis of scarce and partly degraded tissues. We show that very few cells are required for LCM-seq, which implies that it could be of great use in other applications, including cancer biology, pathology and biomarker identification.

## Methods

**Ethics statement.** All work including animal or human tissues was carried out according to the Code of Ethics of the World Medical Association (Declaration of Helsinki). Animal procedures were approved by the Swedish animal ethics review board in Stockholm (Stockholms Norra Djurförsöksetiska nämnd), ethical approval number N82/13. Ethical approval for the use of the human post-mortem samples was obtained from the regional ethical review board in Stockholm, Sweden (Regionala Etikprövningsnämnden, Stockholm, EPN), ethical approval number 2012/111-31/1. Fresh frozen human post-mortem tissues were obtained through the Netherlands Brain Bank (NBB) and the National Disease Research Interchange (NDRI) with the written informed consent from the donors or the next of kin.

**Animal models and tissue dissection.** Mice (B6 and FVB, wild-type, from Jackson Laboratories stock numbers 004435 and 005052) were housed under standard conditions with a 12-h dark/light cycle and had access to food and water *ad libitum*. Five-day-old pups ($n = 7$) (males and females) from three different litters were sacrificed by decapitation for spinal cord dissection. Brains were dissected from 8-week-old females ($n = 3$, two different litters) after anaesthetizing the animals with 2,2,2-Tribromoethanol (Sigma-Aldrich) followed by decapitation. All dissection tools and work surfaces were cleaned with RNaseZAP wipes (Ambion). Tissues were dissected and snap frozen in 2-Methylbutane (Sigma-Aldrich) on dry ice within 15 min after decapitation. Tissue was stored at $-80\,^\circ C$ until further processing.

**Tissue sectioning.** Mouse spinal cords were embedded in pre-cooled OCT embedding medium (Histolab) on dry ice to avoid thawing of the tissue as much as possible. Thin coronal sections (12 μm) were prepared in a cryostat at $-20\,^\circ C$ and placed onto PEN membrane glass slides (Zeiss) that were kept at $-20\,^\circ C$ during the sectioning and subsequently stored at $-80\,^\circ C$ until further processing. For collection of small neurons, mouse brains were sectioned (without embedding) at 5 μm thickness. From human tissues, serial coronal sections of 10 μm thickness from midbrain or spinal cord were prepared as described for mouse tissue while omitting the embedding step.

**Reagents.** For all experiments, nuclease-free water ($H_2O$) and phosphate buffered saline (PBS) (LifeTechnologies) was used and all reagents were of molecular biology/PCR grade if available. Only tubes that were certified nuclease-free were used. All following steps were carried out on work surfaces and with equipment that was cleaned with RNaseZAP.

**Tissue staining procedures before laser capture microscopy.** As MNs are easily identifiable by their distinct location in the ventral horn of the spinal cord and brainstem and by their large soma size, mouse and human tissues were subjected to a quick histological (Nissl) staining based on the Arcturus Histogene Staining Kit protocol[1]. Ethanol solutions of different concentrations for the staining protocol were prepared from 99.7% EtOH Aa (Solveco) and nuclease-free $H_2O$. Slides were kept in a slide box on dry ice for transportation and after 30 s of thawing, they were immediately placed into 75% EtOH for light fixation. After incubation for 30 s in $H_2O$, slides were covered with 200 μl of Histogene staining solution (Arcturus, LifeTechnologies) for 20 or 60 s (mouse and human tissue, respectively). Subsequently, slides were washed for 30 s in $H_2O$ and dehydrated in rising ethanol concentrations (75, 95 and 99.7% EtOH, 30 s each). The whole procedure takes

~6 min. Human midbrain dopamine neurons (mDA) were similarly stained and identified with the histological (Histogene) protocol.

**Rapid antibody staining of mDA neurons.** To minimize RNA degradation, the standard immunohistochemistry procedure for LCM of mDA neurons stained against tyrosine hydroxylase (TH) was kept to ~45 min (ref. 2). Before initiating the staining, the streptavidin-biotin solution (Vectastain ABC kit Elite, Vector laboratories) was mixed in a 15 ml falcon tube that was covered with foil and kept in motion for at least 30 min before use. The membrane slides with the sections were thawed for 1 min and subsequently fixed in cold acetone (Fluka/Sigma-Aldrich) for 5 min. Slides were washed three times in PBS for 1 min and incubated with sheep anti-TH primary antibody (1:25, Pel-Freez, catalogue number P60101-150) in PBS with 0.25% Triton X-100 (Sigma-Aldrich) for 4 min. After three washes with PBS a biotinylated anti-sheep secondary antibody (1:25, Jackson Laboratories, catalogue number 713-065-147) in PBS with 0.25% Triton X-100 was added to the slides and incubated for 4 min. Slides were subsequently washed three times with PBS and incubated with the ABC solution for 4 min, followed by another washing step in PBS. After incubation with 3,3′-diaminobenzidine solution (DAB, Vector laboratories) for 1–2 min, slides were washed with PBS and gradually dehydrated in rising ethanol concentrations ($H_2O$, 50, 75, 95, and 99.7% EtOH, 30 s each). After all staining procedures, slides were air-dried for additional 3 min and immediately subjected to laser microdissection.

It should be noted that quick staining for LCM requires an antibody of excellent quality. Furthermore, to enable a short staining period the antibody is used at a very high concentration, for example, the anti-TH primary antibody in our study was used at 1:25 dilution in LCM, while in regular immunofluorescence staining on fixed tissues it would be used at 1:1,000. Therefore, in addition to conducting our standard staining procedure, as described above, we also performed longer incubations with the sheep anti-TH primary antibody, for 20 or 60 min, to investigate the consequence of increased staining duration on LCM-seq.

**Laser capture microscopy.** Slides were placed into the slide holder of the microscope (Leica DM6000R/CTR6500) and cells were captured using the Leica LMD7000 system. To keep contamination by surrounding cells to a minimum, cutting outlines were drawn closely around individual cells (Supplementary Figs 1c, 4b,c, 6c,g). In spinal cord sections, only cells with an area of more than 200 μm$^2$ and a visible nucleus with nucleolus in the ventral horn of the spinal cord were selected. For the analysis of small neurons, cells were collected from the dorsal motor nucleus of the vagus nerve and the hypoglossal nucleus that had an area of 130–200 μm$^2$ and 200–300 μm$^2$, respectively. mDA neurons from the SNc or ventral tegmental area (VTA) were captured based on their size ($>200$ μm$^2$), the presence of melanin and a visible nucleus. The same groups of mDA neurons were dissected after a quick tyrosine hydroxylase staining on a separate membrane slide with the same tissue distribution. Cells were cut at 40× magnification while keeping laser power to a minimum. Relative humidity for all experiments was between 17 and 65% and temperature range was between 21 and 27 °C. The Leica system uses gravity to collect cells, which fall into the cap of the collection tube (0.2 ml PCR tubes, Biozym Scientific) that is placed directly under the slide with the membrane facing down. After cells were collected in the cap a small volume of lysis buffer (0.2% Triton X-100, with 2 U μl$^{-1}$ recombinant RNase inhibitor, Clontech) was added, followed by pipetting up and down five to ten times. Neurons (~150–200) isolated from human tissues were collected in 5 μl lysis buffer. For mouse MNs, amounts of lysis buffer were reduced according to sample size: for 120 and 50 cells 5 μl of lysis buffer were used, for 30 cells 4 μl and for 10, 5, 2 and 1 cell samples 2.5 μl of lysis buffer were used. Samples were spun down in a table centrifuge (VWR) for 10 s and briefly placed back into the tube holder of the microscope to check that no cells were remaining in the cap. The tubes were carefully sealed with parafilm (Pechiney Plastic Packaging), labelled, and snap frozen on dry ice. The whole process from thawing the slides until sample freezing never took longer than 3 h in any of the experiments. Samples were kept in collection tubes at $-80\,^\circ C$ until further processing. Preparations from empty tubes that were kept in the collector while capturing cells into adjacent PCR tubes served as negative controls. For the mouse study, samples were collected randomly from three to five different animals for each experimental group. The sample size for low cell number groups was increased due to the expected higher technical and biological variation. The exact numbers of samples and animals for each group are stated in Table 1. For human post-mortem tissues, we analysed samples from at least three different individuals for each group. Exact sample sizes are listed in Table 2.

**RNA extraction.** For RNA extraction from LCM dissected MNs the Arcturus PicoPure RNA Isolation Kit was used[1]. One hundred and twenty cells were collected into the cap as described above. Extraction buffer (10 μl) was added to the cap, followed by pipetting up and down ten times, and another 40 μl of extraction buffer were added into the tube. Samples were spun down for 1 min and incubated for 30 min at 42 °C (Thermomixer comfort, Eppendorf). The subsequent steps were carried out according to the manufacturers protocol. Purified RNA was recovered with 12 μl elution buffer. One μl of each sample was run on an RNA 6000 Pico chip on the Agilent 2100 Bioanalyzer to evaluate RNA quality and 5 μl were used for cDNA library preparation.

**Differentiation of mESCs into spinal motor neurons.** Hb9-eGFP mESCs (received from Dr Sebastian Thams and Dr Hynek Wichterle, Columbia University) were differentiated into spinal MNs using a previously published protocol[28], with some modifications. Specifically, embryoid body (EB) formation was induced by plating dissociated mESCs into petri dishes (Sarstedt) in media composed of DMEM/F12 (Invitrogen) and Neurobasal (Invitrogen) (1:1), supplemented with B27 (Invitrogen), Penicillin ($10^4$ U ml$^{-1}$)/Streptomycin ($10^4$ µg ml$^{-1}$) (Invitrogen), 1 mM L-glutamine (Invitrogen) and 0.1 mM β-mercaptoethanol (Invitrogen). Cells were cultured at 37 °C in 5% CO$_2$, while shaking at a speed of 30 r.p.m. to ensure uniform EB formation (ELMI Sky Line Shaker 005-205). After two days of EB formation, MN induction was initiated by adding 100 nM retinoic acid (RA, Sigma) (for caudalization) and 500 nM Smoothened agonist (SAG, Calbiochem) (for ventralization) to the culture media. The patterning towards a spinal motor neuron fate[29] continued for 4 days, after which cells were dissociated using TrypLE Express (Invitrogen) and manually picked for Hb9-eGFP expression and placed into individual tubes containing 4 µl lysis buffer. Sequencing libraries were prepared as described for the LCM-isolated neurons above and subjected to single live cell sequencing[30]. Throughout the culturing period, media change was conducted every 1–2 days.

**cDNA and sequencing library preparation.** Library preparation for sequencing on an Illumina HiSeq2000 sequencer was carried out with a slightly modified version of the Smart-seq2 protocol[11,12] and will therefore be described in detail here. All workbenches and equipment were cleaned with RNaseZAP and DNAoff (Takara). All steps were carried out on ice unless otherwise specified. Reverse transcription (RT) and PCR cycles as well as heat incubation steps were performed in a BioRad T100 Thermal Cycler.

Before RT, samples were incubated with a mix consisting of 1 µl (2 ul for the human tissue samples) 10 mM dNTP mix (all four nucleotides at 10 mM each, Fermentas), 1 µl 10 µM oligo dT primer (Invitrogen, 5′-AAGCAGTGGTATCAAC GCAGAGTACT$_{30}$VN-3′) and 0.1 µl of ERCC spike ins ($2.5 \times 10^5$ dilution from the stock) at 72 °C for 3 min to open the secondary structure of the RNA. Immediately afterwards, samples were placed on ice (snap cooling). The following mix was added to the sample: 2 µl SSRTII 5× buffer, 0.5 µl 100 mM DTT, 0.5 µl 200 U µl$^{-1}$ SSRTII (all LifeTechnologies), 2 µl 5 M betaine (Sigma-Aldrich), 0.1 µl 1 M MgCl$_2$ (Sigma-Aldrich), 0.25 µl 40 U µl$^{-1}$ RNase inhibitor and 0.1 µl 100 µM TSO-LNA-oligo (Exiqon, 5′-AAGCAGTGGTATCAACGCAGAGTACrGrG + G-3′). The RT reaction was carried out as follows: 90 min at 42 °C, then 10 cycles of (2 min at 50 °C and 2 min at 42°), and finally 15 s at 70 °C. Amplification of the cDNA was performed adding KAPA HiFi HotStart Ready mix (12.5 µl of × 2) with 0.2 µl of 10 µM ISPCR primers (Invitrogen, 5′-AAGCAGTGGTATCAACGCAGAGT-3′), and 2.3 µl H$_2$O to the first strand cDNA with the following PCR cycle: 3 min at 98 °C, then 18 cycles of (20 s at 98 °C, 15 s at 67 °C, 6 min at 72 °C), and finally 5 min at 72 °C. The number of PCR cycles was increased to 21 for the analysis of small neurons. After bead purification, the concentration of the cDNA library was measured with an Agilent 2100 Bioanalyzer using the High Sensitvity DNA kit (Agilent). The gained cDNA profile is a valuable quality control.

For the generation of sequencing libraries, the Smart-seq2 protocol uses transposase-based tagmentation to fragment the cDNA and ligate the sequencing adaptors at the same time in a very short reaction. The tagmentation reaction was carried out with 1 ng cDNA, 0.4 µl of in house Tn5 (ref. 31) in 5 µl 40% PEG (Sigma P1458) and 4 µl 5xTAPS-Mg buffer in a final volume of 20 µl for 5 min at 55 °C. Directly afterwards, 5 µl of 0.2% SDS (Sigma-Aldrich) were added, the sample was mixed by pipetting up and down, followed by an incubation step for 5 min at room temperature to strip the enzyme off the cDNA. Samples were placed back on ice and 5 µl of sequencing indices (diluted 1:5 in H$_2$O, Nextera XT Sequencing Index Kit, Illumina) were added. The index ligation and enrichment PCR was performed with 1 µl of 1 U µl$^{-1}$ Kapa HiFi polymerase in 5× Fidelity buffer, 1.5 µl 10 mM dNTP (Kapa Biosystems) in a final volume of 50 µl using the following programme: 3 min at 72 °C, 30 s at 95 °C, then 10 cycles of (10 s at 95 °C, 30 s at 55 °C, 30 s at 72 °C), and finally 5 min at 72 °C. Before pooling the sequencing libraries, a purification step with magnetic beads was performed and the concentration of each sample was determined using a Qubit with the dsDNA high sensitivity kit (LifeTechnologies). An equal amount of cDNA was used to pool up to 30 samples (mouse and human, respectively) that were sequenced in one lane.

**Read mapping and gene expression quantification.** Samples were sequenced using an Illumina HiSeq2000 sequencer and reads were 43 bp in length. RNA-seq reads of mouse samples were mapped to the mouse reference genome mm10 (Ensembl version 78) by employing STAR[32] (version 2.4.1) with parameter --outSAMstrandField intronMotif enabled. For the human samples, RNA-seq reads were mapped to the human reference genome hg38/GRCh38 (Ensembl version 81) using STAR with the same parameter. The number of uniquely mapped reads of each mouse and human sample was calculated using featureCounts[33] (version 1.4.6) with default parameters. To quantify and normalize the expression of genes/transcripts, Cufflinks and Cuffnorm[34] (version 2.2.1) was used with parameter -library-norm-method geometric. Quality control was conducted, the mouse small MN samples (Supplementary Fig. 4) with <1 million reads or <70% mapping ratio to the mm10 genome were removed from the analyses. For the analyses of human samples with different staining time (4, 20 and 60 min) used in Supplementary Fig. 8, we required the samples with at least one million reads and >7,000 genes expressed

≥1 RPKM. For the remaining analyses of corresponding figures, the samples with <1 million reads or <70% mapping ratio to the genome or <7,000 genes expressed ≥1 RPKM were removed. Finally, 95 mouse and 52 human samples with high quality remained for further analysis.

**Gene body coverage assessment.** We investigated the gene body coverage for the mouse MNs subjected to LCM-seq and live single mESC-derived MNs sequenced by Smart-seq2 using RSeQC[35]. We first divided the mouse transcripts (Ensembl version 78) into three different groups according to the length: (i) <3 kb (62.5%); (ii) 3–10 kb (36.3%) and (iii) >10 kb (1.2%). Then we evaluated the 5′ to 3′ coverage bias of these three different groups of transcripts for mouse MN samples using RSeQC with default parameters.

**Clustering based on variable genes.** To perform PCA/heatmap clustering for specific samples based on variable genes in expression, first the function of varianceStabilizingTransformation in DESeq2 (ref. 36) (version 1.6.3) was applied to the matrix that included uniquely mapped read counts of each gene feature. Then PCA/heatmap clustering was performed based on different numbers of top variable genes in expression to optimize the clustering.

**Differential expression calling and gene enrichment analysis.** To compare gene expression differences for human/mouse samples, differential expression analysis with DESeq2 (ref. 36) was conducted. The cutoff for selecting significantly differentially expressed genes was an adjusted $P$ value <0.05 (Wald test). Further, gene ontology (GO) enrichment analysis for the differentially expressed genes was carried out using topGO[37] with algorithm of parentChild. The top 16 enriched biological processes with adjusted $P$ value <0.01 (Fisher's exact test) were selected and shown in the corresponding Supplementary Figures.

**Significance tests.** The statistical analysis of cDNA yield in the mouse study (Supplementary Fig. 1f) was performed on $\log_{10}$-transformed data to meet the assumptions of a one-way ANOVA. One-way ANOVA with Dunnett's *post hoc* analysis for multiple testing correction with the RNA extraction group as a control group (95% confidence interval) was performed using GraphPad Prism 6 and adjusted $P$ values are reported. For comparing the number of detected genes between two different groups of samples (Figs 2c and 4d), Student's $t$-test (one-tailed) was used and mean ± s.e.m. is shown in corresponding bar plots to represent the results.

**Data availability.** All the RNA sequencing data generated in this study have been submitted to the Gene Expression Omnibus (GEO) of the National Center for Biotechnology Information under the accession number GSE76514.

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

## Acknowledgements

This work was funded by grants to E.H. from EU Joint Programme for Neurodegenerative Disease (JPND) (529-2014-7500); Ragnar Söderbergs Stiftelse (M245/11); Swedish Medical Research Council (Vetenskapsrådet) (2011-2651); Karolinska Institutet Delfinansiering av doktorand (KID); Åhlén-stiftelsen (mB8/h11, mB8/h12, mB8/h13, mB8/h14 and mA1/h15), Parkinsonfonden (795/15) and NEURO Sweden; by grants to Q.D. from Swedish Research Council (Vetenskapsrådet) (2014-2870); Svenska Sällskapet för Medicinska Forskning; Jeanssons Stiftelser; Karolinska Institutet Delfinansiering av doktorand (KID); and by grants to R.S. from Swedish Research Council and Swedish Foundation for Strategic Research (FFL4). Human post-mortem tissues were kindly received from the Netherlands Brain Bank (NBB) and the National Disease Research Interchange (NDRI). The authors thank Mattias Karlen for generating the schematics in Fig. 1.

## Author contributions

E.H. and Q.D. conceived the project. S.N., G.C., J.A.B., Q.D. and E.H. designed experiments. S.N., G.C., J.A.B., M.Y., H.S. and M.C. acquired data. S.N., G.C., J.A.B., M.Y., H.S., M.C., R.S., Q.D. and E.H. analysed data. E.H. and Q.D. supervised the project and wrote the manuscript with the help of S.N., G.C., J.A.B. and R.S. All authors edited and gave critical input on the manuscript.

## Additional information

**Accession codes:** All the RNA sequencing data generated in this study have been submitted to the Gene Expression Omnibus (GEO) of the National Center for Biotechnology Information under the accession number GSE76514.

**Competing financial interests:** The authors declare no competing financial interests.

