## [Peer Review File · Nature Communications]

Reviewer #1 (Remarks to the Author):

Hedlund and colleagues developed a new low number of cells RNA-seq technique for fixed tissue sections that coupled laser capture microscopy (LCM) with single cell RNA-seq strategy (LCM-seq). They showed that omitting RNA purification step to run SMART-seq2 directly on the cell lysate works much better than that with RNA purification step. They applied the method to mouse cervical spinal motor neurons (MNs), lumbar spinal MNs, using 120 cells for each sample, to show the gene expression difference between these cells at different location in the spinal cord. The top 20 differentially expressed transcription factors deduced from database AnimalTFDB 2.0 were all detected as differentially expressed by LCM-seq. Then the authors further used LCM-seq method to analyze spinal MNs (6 samples) and midbrain dopamine (mDA) neurons (7 samples) from the human post mortem tissues. They found 4,903 genes were differentially expressed between these two types of cells. Then they applied LCM-seq method to SNc and VTA mDA neurons after either Histogene staining or short antibody staining. And 111 genes were differentially expressed between these two types of mDA neurons. The work is interesting and in general solid. However, although the author claimed single-cell or near-single-cell resolution, majority of the data generated to address biological questions are on 120 cells for each sample. So they do not show the full power of the technique. However, it is still very nice work if they can fully address the following minor revision points.

Minor points:

1. The authors used the amount of PCR product to estimate the effectively amplified mRNAs from different number of cells (120, 50, 30, 10, 5, 2, 1 cells). However, the PCR product is not pure, containing significant amount of non-specific byproducts. Since the authors spiked-in ERCC RNAs into each sample, they should use ERCC to estimate the total copy numbers of mRNAs in each sample to get a much better estimation of the effectively amplified mRNAs in each sample. In this way we will also know usually how many total copy numbers of mRNAs in each sample or single cell are needed for LCM-seq to work robustly. The motor neuron cells are quite large and probably contain abundant amount of mRNAs. I suspect that LCM-seq technique will not work for a single cell of relative small size, for example the diameter less than 10um ones.
2. SMART-seq2 is believed to be able to recover full-length cDNAs from single cells. The authors should analyze the 5' to 3' coverage bias of the mRNAs by LCM-seq method for different lengths of mRNAs (for example, <3kb, 3-10kb, >10kb) and see if in these heavily fixed tissue sections, LCM-seq (SMART-seq2) still can recover full-length cDNAs from majority of them, especially the very long ones (>10kb). If not, the authors need to explain why.
3. Page 6, Line 137: 'The 5, 2, and 1 cell samples showed lower mean numbers of detected genes but were still comparable to a previous study using live single embryonic cells'. This is wrong. Different types of cells can have very different total number of mRNAs within each individual cell. So compare the numbers of genes detected in neurons with those in blastocyst cells is meaningless. If the authors want to claim the high sensitivity of LCM-seq, they should isolate single live cervical MNs and use standard SMART-seq2 to analyze them. And then compare between the LCM-seq on single cell from fixed tissue section and standard SMART-seq2 on live single cells to get the sensitivity of LCM-seq method.
4. When the authors claim that the neurons they got are very pure ones and LCM-seq even permit them to the gene expression difference between neurons at different location in the spinal cord, they should show that these samples are really clean ones. They should get all of the main glial cell marker genes from literature and analyze their LCM-seq data of neurons and see if they really do not detect any of these glial marker genes in their neuron samples.
5. The authors emphasize that 'short' (45min) antibody staining protocol make LCM-seq still work. However, majority of the practical immunostaining protocols need much longer time, especially when also need secondary antibody staining to get the florescence signal amplified. So the authors should show that what is the maximal immunostaining time that the protocol still works for LCM-seq. This information will be greatly helpful for the readers who want to copy and follow their technique.

Reviewer #2 (Remarks to the Author):

Here the authors have combined LCM with RNA-seq from multiple cells down to single neurons and assessed the robustness of the methodology. The work has focussed on the overall detection of transcript numbers, combined with applying the method to a range of sub-anatomical regions in mouse as well as human tissue. Overall the data are well described and the figures are well presented with appropriate statistical tests / PCA analysis. There is certainly an increased interest in the transcriptomics of isolated cell populations, whether by FACS or dissection or single cell technologies (e.g. C1, etc.); thus although there will undoubtedly be many that are keen to keep abreast of the newest approaches. Yet this study falls short of generating significant biological insight into the cell populations under study for such a high-impact general interest journal. There have been many studies using LCM in neurons, some of which are cited here, yet the application of this method to fewer cells - although very valuable - warrants a far more detailed assessment beyond stating that populations cluster based on the 'top 500b variable' transcripts. For example, the Hox gene data are interesting, but cannot be interpreted without a developmental timecourse. In addition, what were their controls for non-neuronal transcripts / controls for the morphology / section bias of selected cells in very small populations? My opinion is that the manuscript in the current form would be highly suited to a methodological journal where it will be highly cited.

Reviewer #3 (Remarks to the Author):

The use of LCM for transcriptome analysis for small numbers of cells has been problematic for exactly the reasons outlined by the authors. As such the technical advances reported by Nichterwitz, et al. are important, novel advances. Furthermore, their extension of the new methodology to discover both expected (hox genes) as well as unexpected differences between different pools of motor neurons is important. Additional analysis of human neurons from post-mortem tissue is also an important technical extension, and the authors report significant differences in different classes of human dopaminergic neurons. I recommend this manuscript for publication after two major criticism are addressed.

1) A promise of full-length transcriptomes, the major purported advance of smart seq2, is the evaluation of splice isoforms. There is no evaluation of this in the current manuscript; yet, there should be.

2) The authors demonstrate single cell success at some level, but there is no evaluation of this data. It was disappointing that all human neurons were analyzed in pools of 100+ cells. Additional analysis of single cells would greatly enhance the impact of this study.

REVIEWERS' COMMENTS:

Reviewer #1 (Remarks to the Author):

The authors have addressed all of my concerns. Just one minor point- I don't agree with the authors to claim that 'Thus, LCM results in the isolation of highly pure motor neuron samples.' (Page #6, Line 127). Based on Supplementary Figure 2a, we can clearly see that some of the 2-cell or even 1-cell samples express glial cell markers (such as GFAP or AIF1) with RPKM higher than 10 (Even though this is many folds lower than in glial cells). So I think the authors should tone down this claim to something like 'Thus, LCM results in the isolation of relatively pure motor neuron samples'.

Rebuttal letter, Page #3, Line #17: 'Reviewer's Fig. 1b-c' should be 'Reviewer's Fig. 2b-c'

Reviewer #2 (Remarks to the Author):

The authors have gone to considerable effort to improve the manuscript, in particular to address the issues regarding specificity / quality of the data from the smallest samples and contamination from non-neuronal cells. These new figures and discussion will greatly enhance the ability of readers to apply the method successfully and to understand the limits of the approach.

Reviewer #3 (Remarks to the Author):

The author's have done an excellent job of addressing my concerns. In particular, comparison of single live ES-derived motor neurons to LCM-SEQ motor neurons is a very compelling demonstration of the veracity of their approach. I recommend this manuscript for publication with no further revisions.

Point-by-point answer:

Reviewer #1:

Hedlund and colleagues developed a new low number of cells RNA-seq technique for fixed tissue sections that coupled laser capture microscopy (LCM) with single cell RNA-seq strategy (LCM-seq). They showed that omitting RNA purification step to run SMART-seq2 directly on the cell lysate works much better than that with RNA purification step. They applied the method to mouse cervical spinal motor neurons (MNs), lumbar spinal MNs, using 120 cells for each sample, to show the gene expression difference between these cells at different location in the spinal cord. The top 20 differentially expressed transcription factors deduced from database AnimalTFDB 2.0 were all detected as differentially expressed by LCM-seq. Then the authors further used LCM-seq method to analyze spinal MNs (6 samples) and midbrain dopamine (mDA) neurons (7 samples) from the human post mortem tissues. They found 4,903 genes were differentially expressed between these two types of cells. Then they applied LCM-seq method to SNc and VTA mDA neurons after either Histogene staining or short antibody staining. And 111 genes were differentially expressed between these two types of mDA neurons. The work is interesting and in general solid. However, although the author claimed single-cell or near-single-cell resolution, majority of the data generated to address biological questions are on 120 cells for each sample. So they do not show the full power of the technique. However, it is still very nice work if they can fully address the following minor revision points.

Response: We sincerely thank the reviewer for the positive comments on our study and for the constructive criticism that has helped us significantly improve our manuscript.

Minor points:

1. The authors used the amount of PCR product to estimate the effectively amplified mRNAs from different number of cells (120, 50, 30, 10, 5, 2, 1 cells). However, the PCR product is not pure, containing significant amount of non-specific byproducts. Since the authors spiked-in ERCC RNAs into each sample, they should use ERCC to estimate the total copy numbers of mRNAs in each sample to get a much better estimation of the effectively amplified mRNAs in each sample. In this way we will also know usually how many total copy numbers of mRNAs in each sample or single cell are needed for LCM-seq to work robustly. The motor neuron cells are quite large and probably contain abundant amount of mRNAs. I suspect that LCM-seq technique will not work for a single cell of relative small size, for example the diameter less than 10um ones.

Response: We agree with the reviewer that it is very valuable to understand what total mRNA copy number is needed for LCM-seq to work robustly. We also appreciate the suggestion of using ERCC spike-ins to calculate the total mRNA copy number. We have conducted the calculation based on the expression of ERCC spike-ins and ERCC molecules added to each sample, see Table 1 below. However, some caveats exist in calculation of RNA molecules using ERCC spike-ins as these are a set of synthetic RNAs added to each sample and their capture efficiency of reverse transcription could be different from native mouse RNAs¹.

Reviewer's Table 1. Number of molecules estimated based on ERCC spike-ins.

Group	No. of samples	Mean number of molecules	SEM
Extract120	5	18,165,200.41	5,766,847.34
120	4	22,896,274.27	2,459,108.77
50	8	14,585,497.52	3,449,087.66
30	9	7,290,153.26	1,061,407.46
10	10	3,229,725.51	370,923.53
5	8	2,089,398.90	343,248.63
2	13	1,006,930.47	158,247.87
1	13	871,162.37	120,306.18
SNs_5	3	312,695.00	148,594.90
SNs_1	5	108,973.49	29,376.71

SNs: small motor neurons

Therefore (as suggested), to further validate our method, we investigated the cell size limit of LCM-seq by isolating motor neurons from the dorsal motor nucleus of the vagus nerve (DMX; areas of $130\text{-}200\mu\text{m}^2 \approx 12.9\text{-}16\mu\text{m}$ diameter) and the hypoglossal nucleus (areas of $200\text{-}300\mu\text{m}^2 \approx 16\text{-}19.5\mu\text{m}$ diameter) that are much smaller in size than spinal motor neurons (areas of $500\text{-}750\mu\text{m}^2 \approx 25.2\text{-}30.9\mu\text{m}$ diameter), which we previously used for our single-cell LCM-seq analysis. We collected single cells and pools of five cells from these two anatomical nuclei, which demonstrates that LCM-seq can indeed work on a single cell level for smaller cells, see Reviewer's Fig. 1a-e. This new data is included in the revised manuscript in the results section on page 7-8 and as Supplementary Fig. 4a-e.

Reviewer's Figure 1. LCM-seq of smaller motor neurons (SNs). (a-c) LCM-seq of SNs, single or pools of 5 cells, collected from the DMX (area of the cells: $130\text{-}200\mu\text{m}^2$) and the hypoglossal nucleus (area of the cells: $200\text{-}300\mu\text{m}^2$). (d-e) Analyses demonstrated that cells with smaller areas can be successfully sequenced using this method. (d) Single cell samples (N=5) contained an average of 4,958 detectable genes, while a mean of 6,945 detectable genes were identified in 5-cell samples (N=3). (e) The percentage of mappable reads were >70% and comparable between single and 5-cell groups (displayed as mean \pm SEM).

2. SMART-seq2 is believed to be able to recover full-length cDNAs from single cells. The authors should analyze the 5' to 3' coverage bias of the mRNAs by LCM-seq method for different lengths of mRNAs (for example,

<3kb, 3-10kb, >10kb) and see if in these heavily fixed tissue sections, LCM-seq (SMART-seq2) still can recover full-length cDNAs from majority of them, especially the very long ones (>10kb). If not, the authors need to explain why.

Response: As suggested by the reviewer we have investigated the 5' to 3' coverage bias of RNAs in mouse motor neurons subjected to LCM-seq. We performed analyses for transcripts with the length of < 3kb (62.5%), 3-10 kb (36.3%) and > 10 kb (1.2%) and compared with live single motor neurons derived from mouse embryonic stem cells (mESCs), as shown below in Reviewer's Fig. 2a-c. This new data set has been included in Supplementary Fig. 3a-c, and in the results section on page 6-7 in the revised manuscript. Live single mESC-derived motor neurons sequenced by Smart-seq2 displayed a better gene body coverage than LCM-seq samples, indicating that the LCM procedure, with its associated tissue dissection, cryostat sectioning and staining, induces, as expected, partial degradation of RNAs. However, live single cells also show a 3' bias in gene body coverage for longer transcript, 3-10 kb, with the Smart-seq2 protocol (Reviewer's Fig. 1b-c).

For LCM coupled with Smart-seq2, both RNA extraction and direct lysis showed a 3' bias for the transcript coverage and the bias was increased for longer transcripts, especially for transcripts > 10 kb. The RNA extraction method showed a slightly better coverage of transcripts <3kb than the direct lysis approach, see Reviewer's Fig. 2a below and Supplementary Fig. 3a and page 6-7 of the results section in the revised manuscript. However, a higher number of genes could be detected using our direct lysis approach (see Fig. 2c in the revised manuscript). This could be due to that the RNA extraction method has a purification step, which can cause a loss of RNAs, but at the same time improves the efficiency of reverse transcription for the remaining transcripts. Thus, using the direct lysis instead of RNA extraction, improves the number of detected genes, but causes a slight decrease in 5' to 3' coverage for transcript with length <3kb. However, while Smart-seq2 can possibly be used for splicing analysis on live cells, we do not recommend investigators to conduct such analyses using LCM-seq until an even better tissue processing procedure has been developed.

Reviewer’s Figure 2. Gene body coverage for transcripts with various ranges of length in different groups of samples. (a, b and c) are the gene body coverage for transcripts with length < 3kb, 3-10 kb and > 10 kb in live single motor neurons (MNs) and different cell numbers of mouse cSC groups, respectively. “Single MNs” refers to live ES-derived single MNs sequenced by Smart-seq2. Extract120 represents the groups of 120 cSC neurons subjected to traditional RNA extraction method, while others are groups with different cell numbers of mouse cSC neurons subjected to direct lysis approach.

3. Page 6, Line 137: 'The 5, 2, and 1 cell samples showed lower mean numbers of detected genes but were still comparable to a previous study using live single embryonic cells. This is wrong. Different types of cells can have very different total number of mRNAs within each individual cell. So compare the numbers of genes detected in neurons with those in blastocyst cells is meaningless. If the authors want to claim the high sensitivity of LCM-seq, they should isolate single live cervical MNs and use standard SMART-seq2 to analyze them. And then compare between the LCM-seq on single cell from fixed tissue section and standard SMART-seq2 on live single cells to get the sensitivity of LCM-seq method.'

Response: We agree with the reviewer that it is more appropriate to compare our LCM-seq data of cervical spinal cord (cSC) motor neurons isolated from mouse tissue with single live cSC motor neurons. We have replaced the blastocyst data with new data in our revised manuscript where we have compared single live cSC motor neurons generated from mESCs² (labeled by a Hb9-eGFP reporter to specifically visualize motor neurons³) (Hb9, n=17; Hb9 is a.k.a. Mnx1) with our single LCM-seq cSC motor neuron data (cSC, n=13), see Reviewer’s Fig. 3a,b. This new data has been incorporated in our revised manuscript in Supplementary Fig. 2 and in the results section on page 6 of the revised manuscript. From this comparison, it is evident that the number of genes

expressed at ≥ 1 RPKM in cSC motor neurons isolated by LCM-seq and live cSC mESC-derived motor neurons subjected to Smart-seq2 are comparable.

Reviewer's Figure 3. The number of genes expressed in cervical spinal cord (cSC) motor neurons isolated by LCM-seq and live cSC mESC-derived motor neurons are comparable. (a) Marker expression profile for 17 single live cSC Hb9 (Mnx1) motor neurons. (b) Comparable number of detected genes (≥ 1 RPKM) were found between LCM-dissected cSC motor neurons (cSC, n=13) and live single Hb9-eGFP mESC-derived cSC motor neurons subjected to Smart-seq2 (Hb9, n=17).

4. When the authors claim that the neurons they got are very pure ones and LCM-seq even permit them to the gene expression difference between neurons at different location in the spinal cord, they should show that these samples are really clean ones. They should get all of the main glial cell marker genes from literature and analyze their LCM-seq data of neurons and see if they really do not detect any of these glial marker genes in their neuron samples.

Response: As suggested by the reviewer we have now analyzed the expression of glial markers in our samples to evaluate the level of glial contamination. Specifically, we downloaded the sequencing raw data of glia samples from a recently published paper⁴ (GEO Series accession number GSE52564) and processed this using the same analysis pipeline as in our paper. Subsequently we analyzed the gene expression of the glial markers *Gfap*, *Mfge8*, *Aif1*, *Cx3cr1*, *Gpr17*, *Itpr2* and *Cnksr3* and the motor neuron markers *Islet-1*, *Islet-2*, *Mnx1*, *Chat*, *Nefh* and *Prph* in our isolated motor neurons with the published glial samples. This new data set clearly demonstrates that the isolated cells express high levels of motor neuron markers and very low levels of glial markers, see Reviewer's Fig. 4. Thus, the LCM results in the isolation of highly pure motor neuron samples. This new data is now also included in the results on page 5-6, Supplementary Fig. 2a and in the discussion on page 13.

Reviewer's Figure 4. Expression profile of motor neuron and glia markers in LCM-isolated motor neurons. Motor neurons isolated by LCM expressed high levels of the motor neuron markers *Islet-1/2*, *Mnx1* (*Hb9*), *Chat*, *Nefh* and *Prph*, while containing low levels of the glial markers *Gfap*, *Mfge8*, *Aif1*, *Cxcr1*, *Gpr17*, *Itpr2* and *Cnksr3*. Glial cells (astrocytes, oligodendrocytes and microglia) showed the opposite expression pattern.

5. The authors emphasize that 'short' (45min) antibody staining protocol make LCM-seq still work. However, majority of the practical immunostaining protocols need much longer time, especially when also need secondary antibody staining to get the florescence signal amplified. So the authors should show that what is the maximal immunostaining time that the protocol still works for LCM-seq. This information will be greatly helpful for the readers who want to copy and follow their technique.

Response: We agree with the reviewer that it is important to address how a longer staining procedure would impact the LCM-seq data. Consequently, we have now compared the number of detected genes in human dopamine neurons subjected to LCM-seq that were stained against an antibody towards tyrosine hydroxylase (TH) for 4 min (according to our standard protocol), with 20 min and 60 min antibody incubation times (while maintaining the secondary antibody for the same length of time). For this experiment we isolated approximately 75 dopamine neurons from each post mortem sample, and each staining group included an N of 3 individuals, and several samples/individual, see new Supplementary Table 4. Importantly, we found that the number of detected genes in the three different "primary antibody staining time" groups were comparable, see Reviewer's Fig. 5a, b. Thus, LCM-seq can be applied to longer staining times. This

new data is now included in the revised manuscript in the results section on page 11 and in Supplementary Fig 8a,b. It should be emphasized that to detect neurons using antibodies in LCM, the antibody should perform very well in regular immunohistochemistry. In addition, the working concentration of the antibody is significantly increased in LCM compared to normal staining procedures on formaldehyde-fixed tissues in order to decrease the time needed to detect the antigen in questions. Thus, in our case, we used the anti-TH antibody at 1:25 dilution for LCM, while we normally use this antibody at 1:1,000 for immunofluorescent analysis on fixed tissue. This is now better clarified in the methods part of the paper, see page 20-21.

Reviewer's Figure 5. Effects of increased antibody staining time on LCM-seq performance for human dopamine neurons. (a) The number of detected genes remained comparable in human dopamine neurons with increasing primary antibody incubation time, which ranged from 4 to 60 min (displayed as mean \pm SEM). (b) Dopamine neuron marker gene expression remained unchanged when the primary antibody incubation time increased from 4, to 20 and 60 min.

Reviewer #2:

Here the authors have combined LCM with RNA-seq from multiple cells down to single neurons and assessed the robustness of the methodology. The work has focused on the overall detection of transcript numbers, combined with applying the method to a range of sub-anatomical regions in mouse as well as human tissue. Overall the data are well described and the figures are well presented with appropriate statistical tests / PCA analysis. There is certainly an increased interest in the transcriptomics of isolated cell populations, whether by FACS or dissection or single cell technologies (e.g. C1, etc.); thus although there will undoubtedly be many that are keen to keep abreast of the newest approaches. Yet this study falls short of generating significant biological insight into the cell populations under study for such a high-impact general interest journal. There have been many studies using LCM in neurons, some of which are cited here, yet the application of this method to fewer cells - although very valuable - warrants a far more detailed assessment beyond stating that populations cluster based on the 'top 500b variable' transcripts. For example, the Hox gene data are interesting, but cannot be interpreted without a developmental time course. In addition, what were their controls for non-neuronal transcripts / controls for the morphology / section bias of selected cells in very small populations? My opinion is that the manuscript in

the current form would be highly suited to a methodological journal where it will be highly cited.

Response: We would like to thank the reviewer for the constructive criticism of our paper. We have now expanded our study to include a very careful analysis of the purity of LCM-isolated motor neurons. Here, we have compared the expression of known motor neuron markers (as before) with markers for astrocytes, oligodendrocytes and microglia, see Reviewer's Fig. 4. To facilitate the interpretation of our analysis we compared our motor neuron data with that of glia samples from a recently published paper⁴ (GEO Series accession number GSE52564). Specifically, we analyzed the gene expression of the glial markers *Gfap*, *Mfge8*, *Aif1*, *Cx3cr1*, *Gpr17*, *Itpr2* and *Cnksr3* and the motor neuron markers *Islet-1*, *Islet-2*, *Mnx1*, *Chat*, *Nefh* and *Prph* in our isolated motor neurons and compared with the published glial samples. This new data set clearly demonstrates that the isolated cells express high levels of motor neuron markers and very low levels of glial markers, see Reviewer's Fig. 4. Thus, the LCM results in the isolation of highly pure motor neuron samples. This new data is now also included in the results on page 6, Supplementary Fig. 2a and in the discussion on page 13. Furthermore, our analysis of pools of 10 cervical motor neurons compared to pools of 120 cervical motor neurons or 120 lumbar motor neurons, clearly demonstrated that 10 cervical motor neurons clustered with 120 cervical motor neuron groups and away from the lumbar. This demonstrates that we can represent the diversity of motor neurons within a segment with 10 neurons alone and suggests that we do not have a bias of the cells we select.

Regarding our analysis of Hox genes; Hox genes specify segment identity along the rostral-caudal axis of the developing embryo and determine if a segment will e.g. form part of the head, thorax⁵ or different levels of the spinal cord; cervical and lumbar levels, which are limb-innervating or thoracic which innervates the abdomen^{2,6,7}. Thus, the expression of Hox genes differs depending on the segmental level of the spinal cord⁷. This gradient of Hox gene expression is established during early development, but is maintained in the adult animal, as we have previously demonstrated⁸. Specifically, cervical levels of the spinal cord express Hox4-Hox8 genes, originally induced by low levels of Fgf (from the presomitic mesoderm), while thoracic and lumbar levels express Hox8-Hox9 and Hox10-Hox13, respectively, induced by a progressively higher level of Fgf8. These distinct Hox genes regulate motor column and pool identities. For example, Hox6 determines cervical lateral motor column (LMC) motor neuron identity, while Hox10 specifies lumbar LMC motor neurons⁹⁻¹¹. At each time the Hox code is active, it confers and thus demonstrates the positional identity of each specific body segment in a highly specialized and pre-determined fashion, and thus does not need to be analyzed along a time line to be informative. Consequently, the Hox gene code is an excellent tool to demonstrate the rostral-caudal level of specific motor neuron populations, as we have previously demonstrated in adult rodent CNS tissues⁸ and that we are again utilizing on postnatal animals in this study.

Reviewer #3:

The use of LCM for transcriptome analysis for small numbers of cells has been problematic for exactly the reasons outlined by the authors. As such the technical advances reported by Nichterwitz, et al. are important, novel advances. Furthermore, their extension of the new methodology to discover both expected (hox genes) as well as unexpected differences between different pools of motor neurons is important. Additional analysis of human neurons from post-mortem tissue is also an important technical extension, and the authors report significant differences in different classes of human dopaminergic neurons. I recommend this manuscript for publication after two major criticism are addressed.

Response: We sincerely thank the reviewer for the positive comments on our study and for the constructive criticism that has helped us significantly improve our manuscript.

1) A promise of full-length transcriptomes, the major purported advance of smart seq2, is the evaluation of splice isoforms. There is no evaluation of this in the current manuscript; yet, there should be.

Response: As suggested, we have analyzed gene body coverage for mouse motor neuron samples subjected to the RNA extraction or direct lysis approaches. We found that both RNA extraction and direct lysis methods have a 3' bias in transcript coverage and this 3' bias is increased for longer transcripts, especially those >3 kb in length - see Supplementary Fig. 3/Reviewer's Fig. 2. We also examined the number of detected isoforms between the RNA extraction and direct lysis approaches as well as live single mESC-derived motor neurons. As shown in the Reviewer's Fig. 6, the RNA extraction and direct lysis performed similarly in the number of detected isoforms, while in live single motor neurons a significantly larger number of splice isoforms were detected. It should be noted that the detection/analysis of splice isoforms is sensitive for the gene body coverage and the 3' bias can increase uncertainties of the results. The 3' bias could be caused both by the limitation of Smart-seq2 (as this technology utilizes the poly-A tail for amplification, there will always be a slight 3' bias, especially for longer transcripts) and particularly by RNA degradation due to LCM tissue processing, as is seen in the comparison of the live single motor neurons with LCM-seq motor neurons in Reviewer's Fig. 2. Thus, while Smart-seq2 can possibly be used for splicing analysis on live cells, we do not recommend investigators to conduct splice isoform analysis using LCM-seq at this time.

Although the RNA extraction method showed a slightly better coverage at the 5' of transcripts than our direct lysis approach, a higher number of genes could be detected using direct lysis, see Fig. 2c in the revised manuscript. This could be due to that RNA extraction method contains a purification step, unlike the direct lysis method, which could result in loss of transcripts. However, purified RNA may have a better efficiency of reverse transcription than directly lysed samples and thus a better 5' coverage. We have further discussed the points raised above in our revised manuscript, see page 6-7.

Reviewer's Figure 6. Comparison of the number of detected splice isoforms in different groups of mouse cSC samples. Hb9 represents single live mESC-derived motor neurons, while Extract120 represents the group of 120 cSC neurons subjected to traditional RNA extraction. All remaining groups represent different cell numbers of cSC neurons subjected to direct lysis prior to sequencing (displayed as mean \pm SEM).

2) The authors demonstrate single cell success at some level, but there is no evaluation of this data. It was disappointing that all human neurons were analyzed in pools of 100+ cells. Additional analysis of single cells would greatly enhance the impact of this study.

Response: As suggested by the reviewer we have performed LCM-seq on a lower number of human spinal motor neurons, ranging from 10 cells down to single cells, isolated from post mortem tissues. This analysis demonstrates that LCM-seq can be used successfully on single human motor neurons, see the new Fig. 5 in the revised manuscript and below as Reviewer's Fig. 7. Specifically, human spinal motor neuron subjected to LCM-seq expressed the motor neuron markers *ISLET-1/2*, *MNX1*, *CHAT*, *NEFH* and *PRPH* (Reviewer's Fig. 7a). A large number of detectable genes were identified in all cell groups, but the 1 cell group had fewer detectable genes (Reviewer's Fig. 7b), similar to our data on mouse spinal motor neurons (Figure 2 of the manuscript). Human motor neuron samples also showed a good correlation, with Spearman's correlation of >0.6 for bulk (>100 cells) down to single cell samples (Reviewer's Fig. 7c). PCA clustering based on top 500 variable genes demonstrated that human spinal motor neuron groups clustered well together and away from the human dopamine neurons (Reviewer's Fig. 7d). These new data are included in the revised manuscript in a new Fig. 5, and in the results section on page 12-13.

Reviewer's Figure 7. LCM-seq is applicable on single human motor neurons isolated from post mortem tissue. (a) Human motor neurons expressed the motor neuron-specific markers *ISLET1/2*, *MNX1*, *CHAT*, *NEFH* and *PRPH*. (b) Mean number of genes detected in the distinct groups of different cell numbers; bulk (179.7±30.8 cells), 10, 5 and 1 motor neuron (displayed as mean ± SEM). (c) Gene expression correlation for bulk, 10, 5 and 1 cell samples (Spearman's correlation; genes expressed at ≥ 1 RPKM in at least one sample were used). (d) PCA of bulk, 10, 5 and 1 motor neuron groups and bulk dopamine neurons, based on the top 500 variable genes.

References

1. Stegle, O., Teichmann, S.A. & Marioni, J.C. Computational and analytical challenges in single-cell transcriptomics. *Nat Rev Genet* **16**, 133-145 (2015).
2. Allodi, I. & Hedlund, E. Directed midbrain and spinal cord neurogenesis from pluripotent stem cells to model development and disease in a dish. *Frontiers in neuroscience* **8**, 109 (2014).
3. Wichterle, H., Lieberam, I., Porter, J.A. & Jessell, T.M. Directed differentiation of embryonic stem cells into motor neurons. *Cell* **110**, 385-397 (2002).

4. Zhang, Y. et al. An RNA-sequencing transcriptome and splicing database of glia, neurons, and vascular cells of the cerebral cortex. *The Journal of neuroscience : the official journal of the Society for Neuroscience* **34**, 11929-11947 (2014).
5. Myers, P. Hox genes in development: The Hox Code. *Nature Education* **1**, 2 (2008).
6. Hedlund, E., Karsten, S.L., Kudo, L., Geschwind, D.H. & Carpenter, E.M. Identification of a Hoxd10-regulated transcriptional network and combinatorial interactions with Hoxa10 during spinal cord development. *J Neurosci Res* **75**, 307-319 (2004).
7. Carpenter, E.M. Hox genes and spinal cord development. *Dev Neurosci* **24**, 24-34 (2002).
8. Hedlund, E., Karlsson, M., Osborn, T., Ludwig, W. & Isacson, O. Global gene expression profiling of somatic motor neuron populations with different vulnerability identify molecules and pathways of degeneration and protection. *Brain* **133**, 2313-2330 (2010).
9. Dasen, J.S., Liu, J.P. & Jessell, T.M. Motor neuron columnar fate imposed by sequential phases of Hox-c activity. *Nature* **425**, 926-933 (2003).
10. Shah, V., Drill, E. & Lance-Jones, C. Ectopic expression of Hoxd10 in thoracic spinal segments induces motoneurons with a lumbosacral molecular profile and axon projections to the limb. *Developmental dynamics : an official publication of the American Association of Anatomists* **231**, 43-56 (2004).
11. Wu, Y., Wang, G., Scott, S.A. & Capecchi, M.R. Hoxc10 and Hoxd10 regulate mouse columnar, divisional and motor pool identity of lumbar motoneurons. *Development* **135**, 171-182 (2008).

Point-by-point answers:

We would like to again thank the reviewers for the constructive criticism and insightful comments, which helped us improve our manuscript.

REVIEWERS' COMMENTS:

Reviewer #1 (Remarks to the Author):

The authors have addressed all of my concerns. Just one minor point- I don't agree with the authors to claim that 'Thus, LCM results in the isolation of highly pure motor neuron samples.' (Page #6, Line 127). Based on Supplementary Figure 2a, we can clearly see that some of the 2-cell or even 1-cell samples express glial cell markers (such as GFAP or AIF1) with RPKM higher than 10 (Even though this is many folds lower than in glial cells). So I think the authors should tone down this claim to something like 'Thus, LCM results in the isolation of relatively pure motor neuron samples'. Rebuttal letter, Page #3, Line #17: 'Reviewer's Fig. 1b-c' should be 'Reviewer's Fig. 2b-c'

Response: We sincerely thank the reviewer for the positive comments on our study. We have now addressed the reviewer's concern and have modified our statement to instead say: "Thus, LCM results in the isolation of highly enriched motor neuron samples."

Reviewer #2 (Remarks to the Author):

The authors have gone to considerable effort to improve the manuscript, in particular to address the issues regarding specificity / quality of the data from the smallest samples and contamination from non-neuronal cells. These new figures and discussion will greatly enhance the ability of readers to apply the method successfully and to understand the limits of the approach.

Response: We sincerely thank the reviewer for the positive comments on our study.

Reviewer #3 (Remarks to the Author):

The author's have done an excellent job of addressing my concerns. In particular, comparison of single live ES-derived motor neurons to LCM-SEQ motor neurons is a very compelling demonstration of the veracity of their approach. I recommend this manuscript for publication with no further revisions.

Response: We sincerely thank the reviewer for the positive comments on our study.